# NOX4 Deficiency Exacerbates the Impairment of Cystatin C-Dependent Hippocampal Neurogenesis by a Chronic High Fat Diet

**DOI:** 10.3390/genes11050567

**Published:** 2020-05-19

**Authors:** Piyanart Jiranugrom, Ik Dong Yoo, Min Woo Park, Ji Hwan Ryu, Jong-Seok Moon, Sun Shin Yi

**Affiliations:** 1Department of Biomedical Laboratory Science, College of Medical Sciences, Soonchunhyang University, Asan 31538, Korea; piyanart.jiranugrom@gmail.com; 2Department of Chemical Engineering, King Mongkut’s University of Technology Thonburi, Bangkok 10140, Thailand; 3Department of Nuclear Medicine, Soonchunhyang Hospital-Cheonan, Cheonan 31151, Korea; 92132@schmc.ac.kr; 4Department of Integrated Biomedical Science, Soonchunhyang Institute of Medi-Bio Science (SIMS), Soonchunhyang University, Cheonan 31151, Korea; pmw0269@nate.com; 5Severance Biomedical Science Institute, Yonsei University College of Medicine, Seoul 03722, Korea; yjh@yuhs.ac

**Keywords:** NOX4, hippocampus, neurogenesis, Cystatin C, high fat diet

## Abstract

Hippocampal neurogenesis is linked with a cognitive process under a normal physiological condition including learning, memory, pattern separation, and cognitive flexibility. Hippocampal neurogenesis is altered by multiple factors such as the systemic metabolic changes. NADPH oxidase 4 (NOX4) has been implicated in the regulation of brain function. While the role of NOX4 plays in the brain, the mechanism by which NOX4 regulates hippocampal neurogenesis under metabolic stress is unclear. In this case, we show that NOX4 deficiency exacerbates the impairment of hippocampal neurogenesis by inhibiting neuronal maturation by a chronic high fat diet (HFD). NOX4 deficiency resulted in less hippocampal neurogenesis by decreasing doublecortin (DCX)-positive neuroblasts, a neuronal differentiation marker, and their branched-dendrites. Notably, NOX4 deficiency exacerbates the impairment of hippocampal neurogenesis by chronic HFD. Moreover, NOX4 deficiency had a significant reduction of Cystatin C levels, which is critical for hippocampal neurogenesis, under chronic HFD as well as normal chow (NC) diet. Furthermore, the reduction of Cystatin C levels was correlated with the impairment of hippocampal neurogenesis in NOX4 deficient and wild-type (WT) mice under chronic HFD. Our results suggest that NOX4 regulates the impairment of Cystatin C-dependent hippocampal neurogenesis under chronic HFD.

## 1. Introduction

Hippocampal neurogenesis is critical for learning and memory function in the brain. Hippocampal neurogenesis is regulated by cell proliferation, neuronal differentiation, and cell survival in the granular cell layer (GCL) of the dentate gyrus (DG) and the adjacent sub-granular zone (SGZ) [1]. These three critical components of neurogenesis can be modulated by multiple factors [2,3]. Both adult neurogenesis and developmental neurogenesis are tightly controlled by genetic and molecular programs, which grant a seemingly identical maturation process to adult-born neurons and neurons formed during embryogenesis. This explains the basis of the cytological organization seen in neuronal tissue [2,3]. On the other hand, there are regulatory intrinsic or extrinsic factors that could increase or decrease neurogenesis. These factors can depend on human behavior or the environmental factor and can increase or decrease the formation of new neurons during adulthood [2]. They are critical for understanding the relevance of adult neurogenesis on a functional basis, either behaviorally, cognitively, or clinically [2].

The function of hippocampus is linked to systemic metabolism in the human body via various aspects including the control of behavior for food, nutrient uptake in the gut, thermogenesis, and the interaction with metabolic tissues such as liver, adipose tissue, and skeletal muscle [4,5,6,7,8]. In addition, the function of the hippocampus is affected by the changes of body metabolism via systemic inflammation and hormonal imbalance in human metabolic diseases such as obesity and Type 2 diabetes. Previous studies suggest that the dysfunction of neurogenesis is associated with human metabolic diseases [9,10,11,12]. Hippocampal neurogenesis is suppressed by the increase of oxidative stress and the decrease of a brain-derived neurotrophic factor (BDNF), which is involved in the enhanced hippocampal neurogenesis under a high fat diet (HFD) related to obesity and Type 2 diabetes [13,14,15]. Obesity is linked to the development of mild cognitive impairment and late-life dementia or Alzheimer’s disease [16,17,18]. In addition, diabetes impairs hippocampus-dependent memory, perforant path synaptic plasticity, and adult neurogenesis by reducing hippocampal function via glucocorticoid-mediated effects on new and mature neurons [19]. Although it has been shown that the hippocampus is part of a neural circuit involved with reward and energy regulation [13,14], the understanding of the mechanism or key factors in the regulation of hippocampal neurogenesis under chronic metabolic stress by HFD is still unknown.

Cystatin C is an inhibitor of cysteine proteinases [20]. The function of Cystatin C has been identified in kidney diseases, cardiovascular disease, and neurologic disorders [21,22,23,24,25,26]. The mutation of Cystatin C is linked to the Icelandic type of hereditary cerebral amyloid angiopathy, which is a condition predisposing to an intracerebral hemorrhage, stroke, and dementia [25,26]. Recent studies showed that Cystatin C binds amyloid β and reduces its aggregation and deposition. These results suggest that Cystatin C could be a potential target in Alzheimer’s disease [27,28]. Other studies showed that Cystatin C has a role as a susceptibility gene for Alzheimer’s disease [29,30]. The study for function of Cystatin C in Alzheimer’s disease remains unclear.

NADPH oxidase 4 (NOX4) is an enzyme in seven members of the NADPH oxidase (NOX) family (Nox1-5 and Duox 1 and 2), which is critical for the production of reactive oxygen species (ROS). In the brain, NOX4 is predominantly localized in endothelial cells and neurons in mice and humans [31,32]. Additionally, NOX4 is expressed abundantly in the vascular smooth muscle, endothelial cells, heart, and kidney [33,34,35]. NOX4 plays a role in cellular differentiation and proliferation in heart and muscle [36]. Recently, the role of NOX4 has been identified in neuronal cells in the brain [37,38,39,40]. NOX4 is linked to the regulation of differentiation and survival of neural stem cells and the loss of function of brain pericytes during acute brain ischemia [41]. Moreover, NOX4 is associated with regulating cellular metabolism related to glucose metabolism and mitochondrial metabolism in the cardiac system and cancer cells [42,43]. We have previously reported that NOX4 is linked to fatty acid metabolism via mitochondrial fatty acid oxidation in macrophages [44]. Although the functions of NOX4 were identified in various types of cells, the effects of NOX4 deficiency in the brain under chronic metabolic stress such as obesity or HFD are not well known. Currently, the roles of NOX4 in regulating hippocampal neurogenesis under chronic HFD are unclear.

In this study, we demonstrated that NOX4 deficiency exacerbates the impairment of Cystatin C-dependent hippocampal neurogenesis by chronic HFD. Our results showed that NOX4 deficiency had higher reduction of hippocampal neuronal maturation by chronic HFD. Moreover, we found that NOX4 deficiency had less weight gain by chronic HFD, whereas the food intake was unchanged. Furthermore, NOX4 deficiency resulted in less production of Cystatin C, which is critical for the protection of neuronal function in the hippocampus under chronic HFD. Our results suggest that NOX4 regulates the impairment of Cystatin C-dependent hippocampal neurogenesis under chronic HFD.

## 2. Materials and Methods

### 2.1. Animal Studies

All mouse experimental protocols were approved by the Institutional Animal Care and Use Committee of Soonchunhyang University (protocol #SCH19-0027). C57BL/6J mice (*n* = 8, male, 22~25 g) for wild-type (WT) colony were purchased from the Saeron Bio (Gunpo, Korea). The *Nox4*^−/−^ mice were from the Jackson laboratory (Jackson laboratory, Farmington, CT, USA). *Nox4*^−/−^ mice were previously described [45]. The *Nox4*^−/−^ mice were genotyped using standard PCR using DNA from tail. WT and NOX4 knock out (KO) mice (*n* = 8, male, 22~25 g), which were C57BL/6J strain background were housed at room temperature (22 ± 2 °C) with 60% humidity under a 12-h light:dark cycle (light cycle:dark cycle from 07:00 to 19:00). They had no special disease, pathogen, or genetic defects. They were provided free access to a normal chow (NC) diet (2018S, Harlan, USA) or high fat (HF) diet (Research Diets, New Brunswick, NJ, USA) and distilled water. Mice were used in this study after one week of acclimation. The mice were four grouped into wild-type (WT) (normal chow (NC)/WT) and NOX4 KO (NC/KO) mice that consume the NC diet, and WT (HF/WT) and NOX4 KO (high fat (HF)/KO) mice that consume the HF diet. Body weight and food intake volume every day for 7 weeks were measured in the mice. The animals were sacrificed. Then abdominal and epididymal white adipose tissue (WAT, *n* = 4 each group) were completely removed and weighed. The brains were quickly removed from the cranial cavity, and isolated by hippocampus. The other animals (*n* = 4, each group) were deeply anesthetized with Urethane (1 g/kg; Sigma-Aldrich, St. Louis, MO, USA) and perfused with 0.1 M phosphate-buffered saline (PBS, pH 7.4), as described in the previous study [46].

### 2.2. Analysis of Fat Volume and Total Body Weight 

Mice (4 mice per group) were performed perfusion in each group. Epididymal white adipose tissues (eWAT) were used for the analysis of changes of fat mass by weight gain and cellularity measurement, as described in the previous study [46,47]. Epididymal fat pads were carefully removed without damaging the testicular blood supply, as previously described [48,49]. eWAT were completely separated from the epididymal fat pads and were calculated in each mouse before sacrifice. For objective evidence of the above values, the dorsal views of the animals were photographed during the pericardial phase and the fat tissue with the abdominal organs was easily exposed. Using a ruler, the differences between different groups are easily expressed in the picture.

### 2.3. Reagents and Antibodies

The following antibodies were used: polyclonal rabbit anti-Ki67 (1:1000, Abcam, Cambridge, UK), polyclonal goat anti-doublecortin (anti-DCX) antibody (1:500, Santa Cruz Biotechnology, Dallas, TX, USA), biotinylated rabbit anti-goat IgG (diluted 1:200, Vector Laboratories, Burlingame, CA, USA), and biotinylated goat anti-rabbit IgG (diluted 1:200, Vector Laboratories). Sections were then visualized by staining with 3, 3′-daminobenzidine in 0.1 M Tris-HCl buffer (pH 7.2). Sections were mounted onto gelatin-coated slides with Canada Balsam (Wako, Tokyo, Japan) following dehydration.

### 2.4. Immunohistochemistry and Immunofluorescence Analysis

For immunohistochemistry analysis, brain tissues were removed from the cranial cavities and post-fixed overnight in 4% paraformaldehyde under the temperature at 4 °C, and sectioned at a thickness of 30 μm between −1.82 and −2.46 mm posterior to the Bregma with references to a mouse brain atlas [50]. The sections were stained with an antibody against specific targets. The secondary antibody was biotinylated goat anti-rabbit IgG (Vector Laboratories), and biotinylated rabbit anti-goat (Vector Laboratories) for and neuroblast (doublecortin, DCX), respectively. Subsequently, streptavidin peroxidase complex (Vector Laboratories) was biotinylated for 2 h at 25 °C. Stained brain sections were analyzed by Olympus BX53M microscope and quantified by using Olympus Stream software and ImageJ software v1.52a (Bethesda, MD, USA). To ensure objectivity, all measurements were performed under blinded conditions by two observers per experiment under identical conditions. DCX positive cells were counted. Particularly, the total numbers of neuroblasts and the neuroblasts having tertiary branched dendrites at the SGZ in the DG were analyzed.

### 2.5. Cystatin C and Cytokine Analysis

For Cystatin C and cytokine analysis, the hippocampus in brain from mice were isolated and lysed in Tissue Extraction Reagent I (FNN0071, Invitrogen, Carlsbad, CA, USA). The lysates were centrifuged at 15,300× *g* for 10 min at 4 °C, and the supernatants were obtained. The protein concentrations of the supernatants were determined by applying the Bradford assay (500-0006, Bio-Rad Laboratories (Hercules, CA, USA)). The amount of 100-μg protein lysates from the hippocampus of mice was used for Cystatin C and 110 soluble proteins including cytokines, chemokines, and growth factors analysis by the Mouse XL Cytokine Array Kit (ARY028, R&D systems (Minneapolis, MN, USA)), according to the manufacturer’s instructions. The protein lysates from the hippocampus were incubated with four nitrocellulose membranes containing 111 different captured antibodies printed in duplicate for 16 h at 4 °C. Nitrocellulose membranes were incubated with a detection antibody diluted in an assay buffer for 2 h at room temperature and then incubated with the streptavidin-horseradish peroxidase (HRP) in an assay buffer for 0.5 h at room temperature. The immunoreactive spots on nitrocellulose membranes were detected by the chemical reagent mix and then exposed to X-ray film. Multiple exposure times were used. The pixel densities from positive signals on developed X-ray film were collected and analyzed using a transmission mode scanner and image analysis software (HLImage++ Version 25.0.0r, https://www.wvision.com/QuickSpots.html, Western Vision Software, Salt Lake City, UT, USA). The pixel densities were quantified and determined the relative change in analyte levels. The values of all analytes are shown in Appendix A.

### 2.6. Statistical Analysis

All data are represented as mean ± standard deviation (SD) or standard error of the mean (SEM). All statistical tests were analyzed using a two-tailed Student’s *t*-test for comparison of two groups, and analysis of variance (ANOVA) (with post hoc comparisons using Dunnett’s test) using a statistical software package (GraphPad Prism version 8.0, GraphPad Software Inc. (San Diego, CA, USA)) for comparison of multiple groups. *p* values (*, *p* < 0.05, **, *p* < 0.005, ***, *p* < 0.0005) were considered statistically significant.

## 3. Results

### 3.1. NOX4 Deficiency Decreases Fat Accumulation in Epidermal White Adipose Tissue during Chronic HFD

To investigate the role of NOX4 in hippocampal neurogenesis under chronic HFD, we examined whether genetic deficiency of NOX4 could affect the impairment of hippocampal neurogenesis by chronic HFD. We used a mouse model of chronic HFD for seven weeks. We first analyzed the appearance of mice under HFD or NC diet including the dorsal view, incision of the abdomen (ventral view), and the size of epidydimal fat from the abdomen (Figure 1A). In the dorsal view, *Nox4*^−/−^ mice showed leaner body shape compared to wild-type (WT) mice under HFD (Figure 1A). In ventral view, *Nox4*^−/−^ mice had less internal fat relative to WT mice under HFD (Figure 1A). We next examined whether NOX4 deficiency could reduce fat accumulation in epididymal white adipose tissue (eWAT) under an HFD. We analyzed the volume of eWAT in mice (Figure 1B). *Nox4*^−/−^ mice showed significant less volume of eWAT per body weight relative to WT mice under an HFD (Figure 1B). In contrast, the difference in volume of eWAT per body weight was comparable between *Nox4*^−/−^ mice and WT mice under NC (Figure 1B). These results suggest that NOX4 deficiency decreases systemic fat accumulation under chronic HFD.

### 3.2. NOX4 Deficiency Affects Weight Gain during Chronic HFD

Next, we investigated whether NOX4 deficiency could affect the regulation of systemic metabolism during chronic HFD. We measured the change of weight gain of mice during HFD or NC. The body weight and food intake were measured at the same time every day. *Nox4*^−/−^ mice showed substantial reduction of weight gain compared to WT during HFD (Figure 2A). There is a small difference in weight gain between *Nox4*^−/−^ and WT mice during NC (Figure 2A). We next examined whether NOX4 deficiency could affect food intake during weight gain in HFD. We analyzed the change of food intake between *Nox4*^−/−^ and WT mice during HFD or NC (Figure 2B). In contrast to weight gain, the change of food intake was comparable between *Nox4*^−/−^ and WT mice (Figure 2B). Consistently, there was no difference in the expression of mitochondrial oxidative phosphorylation complex enzymes, which regulate mitochondrial-dependent cellular oxygen consumption and ATP production between *Nox4*^−/−^ and WT mice (Appendix A). These results suggest that the NOX4 deficiency contributes to the alteration of systemic metabolism during HFD.

### 3.3. NOX4 Deficiency Exacerbates the Impairment of Hippocampal Neurogenesis by Chronic HFD

To investigate the role of NOX4 in the impairment of hippocampal neurogenesis by chronic HFD, we examined whether NOX4 deficiency could affect the hippocampal neurogenesis during chronic HFD. We analyzed hippocampal neurogenesis by measuring a DCX-positive neuroblast, which is a neuronal differentiation marker, in *Nox4*^−/−^ and WT mice. Hippocampal neurogenesis drives the development of neuroblast in the SGZ hippocampal DG [51,52]. These neuroblasts migrate into GCL during maturation [50,51]. Since dendrites stretching out from neuroblast signal transmission by associating with the surrounding cerebral parenchymal structure, the number of branches of dendrites as well as the number of neuroblasts is considered [51,52]. We measured the number of DCX-positive neuroblasts, which are differentiated neuroblasts, and the degree of dendrite branching in DCX-positive neuroblasts distributed along the SGZ and GCL of hippocampal DG in *Nox4*^−/−^ and WT mice (Figure 3A). The number of DCX-positive cells and branched dendrites in the SGZ and GCL of hippocampus DG was suppressed in WT mice with HFD relative to NC (Figure 3B,C). Notably, *Nox4*^−/−^ mice showed a significant reduction of DCX-positive neuroblasts and dendrites in the SGZ and GCL of hippocampal DG under HFD when compared to WT mice (Figure 3B,C). Similarly, *Nox4*^−/−^ mice had lower DCX-positive neuroblasts and dendrites in the SGZ and GCL hippocampal DG than WT mice under NC (Figure 3B,C). To investigate the functional morphology of the neuroblast, we analyzed the number of tertiary branched dendrites in neuroblasts by quantifying the sub-branch (Figure 3D). *Nox4*^−/−^ mice resulted in less tertiary branches in dendrites when compared to WT mice under HFD or NC (Figure 3D). Particularly, *Nox4*^−/−^ mice showed higher immature dendrites that have fewer and shorter branches in the middle region of DG under HFD or NC than WT mice (black head arrows in Figure 3B). These results suggest that NOX4 deficiency exacerbates the impairment of hippocampal neurogenesis via neuronal maturation by chronic HFD.

### 3.4. NOX4 Deficiency Suppresses the Production of Cystatin C in Hippocampus during Chronic HFD

We investigated the molecular mechanism by which NOX4 regulates hippocampal neurogenesis under HFD. We examined whether NOX4 could affect the production of neuronal cytokines or chemokines in the hippocampus under HFD. We analyzed the levels of 111 soluble proteins including cytokines, chemokine, and growth factors in the hippocampus from *Nox4*^−/−^ and WT mice with HFD or NC (Figure 4 and Appendix A). Notably, *Nox4*^−/−^ hippocampus had significantly fewer Cystatin C levels during HFD relative to WT mice (Figure 4). Consistently, *Nox4*^−/−^ hippocampus had less Cystatin C production in NC as a basal condition (Figure 4). The changes of the Cystatin C levels were co-related with hippocampal neurogenesis in *Nox4*^−/−^ and WT mice. On the other hand, FGF-1, which is a growth factor and signaling protein encoded by the *FGF1* gene, and IL-28B (a cytokine encoded by the interferon lamda 3 (*IFNL3*) gene) levels in hippocampus were changed in *Nox4*^−/−^ and WT with HFD or NC, whereas these changes were not co-related with hippocampal neurogenesis in *Nox4*^−/−^ and WT mice (Appendix A). These results suggest that NOX4 deficiency suppresses the production of Cystatin C in the hippocampus during chronic HFD.

## 4. Discussion

In this study, we demonstrate that NOX4 regulates the impairment of Cystatin C-dependent hippocampal neurogenesis by chronic HFD. Our results suggest that NOX4 could be a critical molecule for neuronal protection against the impairment of Cystatin C-dependent hippocampal neuronal maturation by chronic HFD.

Imbalanced foods containing high fat or high calories have been implicated in the changes of brain function in human metabolic diseases such as obesity and type 2 diabetes [7,8,9,10]. In addition, maintaining a healthy brain function with increased longevity is an important factor in quality of life [53]. The continuous and active neurogenesis of the hippocampus is a critical requirement for maintaining cognitive and memory capabilities and is also an indicator of the state of brain health. Since hippocampus is also known to be a very pivotal part of the constancy of energy metabolism, the correlation between energy utilization and hippocampal neurogenesis was highlighted recently. Cell proliferation and neuronal differentiation are reduced during inflammation in the SGZ of an animal hippocampus whose weight has been significantly increased by HFD [54]. In addition, previous studies suggested that metabolic stress by obesity and type 2 diabetes is linked to a neuronal disorder such as Parkinson’s disease (PD) and Alzheimer’s disease (AD). Based on our results and other studies, metabolic stress by HFD causes the impairment of hippocampal neurogenesis in a mouse model. Our results show that the number of neuroblast (DCX)-positive cells in *Nox4*^−/−^ mice under HFD as well as dendrites stretching out from those neuroblasts was significantly reduced when compared to WT mice under HFD as well as *Nox4*^−/−^ and WT mice under NC. Our findings suggest that NOX4 deficiency exacerbates the impairment of hippocampal neurogenesis under HFD. Our results are thought to be consistent with previous reports showing a reduction in the subventricular zone (SVZ) of lateral ventricular neurogenesis in a NOX4-deficient condition [55,56]. Similarly, a previous study showed that NOX4 deficiency contributes to the reduction of bone formation by osteogenic impairment [57]. Our findings suggest that NOX4 deficiency exacerbates the impairment of hippocampal neurogenesis via metabolic stress by HFD. Although we showed the role of NOX4 in hippocampal neurogenesis during HFD, there is a limitation to support whether NOX4 can directly affect neuronal differentiation in hippocampal neurogenesis in our current study. Further study for the role of NOX4 on hippocampal neuronal differentiation would need to be studied.

Additionally, our results found that NOX4 deficiency had resistance against the rapid weight gain by HFD. However, there was no difference in food intake and the expression of mitochondrial oxidative phosphorylation complex enzymes, which regulate mitochondrial-dependent cellular oxygen consumption and ATP production between *Nox4*^−/−^ and WT mice. Since the function of NOX4 is linked to the regulation of mitochondrial fatty acid oxidation in our previous study [44], our results suggest that NOX4 could contribute to the regulation of mitochondrial fatty acid oxidation in metabolic organs and cells during HFD. Additionally, further study for the role of the NOX4 gene on hippocampus in the regulation of systemic metabolic regulation, such as fatty acid metabolism, would need to be studied.

Cystatin C plays a neuroprotective role in neurodegenerative diseases including PD, AD, amyotrophic lateral sclerosis (ALS), and subarachnoid hemorrhage (SAH) [58,59,60]. In neurogenesis, the role of Cystatin C is identified in hippocampus [61,62]. The Cystatin C levels are co-related with the prominent neurogenesis [63,64]. In addition, the basal level of neurogenesis in the sub-granular layer of DG are reduced in Cystatin C knockout mice, which support a role for Cystatin C in neurogenesis [63,64]. During acute hippocampal injury or status epilepticus induced epileptogenesis, the Cystatin C gene and protein expression levels are increased in the hippocampus and in the DG [61]. Additionally, a previous study reported that Cystatin C has an important role in the generation of neuronal stem cells from embryonic stem (ES) cells [65]. Consistent with our results, there is an interaction between the levels of hippocampal neuronal maturation and Cystatin C levels in the hippocampus of the NOX4 deficient mice. Our results showed that NOX4 deficiency exacerbates the reduction of Cystatin C production in the hippocampus under chronic HFD compared to WT. These results suggest that the reduction of Cystatin C could be a critical molecular mechanism in the impairment of hippocampal neurogenesis by NOX4 deficiency under chronic HFD. In addition, our results suggest that NOX4 could have an important role in the neuronal protection through hippocampal neurogenesis under human metabolic diseases such as obesity and type 2 diabetes.

Although our results showed the role of NOX4 in hippocampal neurogenesis and Cystatin C production under chronic HFD, there is a limitation to determine the direct or indirect interaction between NOX4 and the regulation of Cystatin C production in hippocampal neurogenesis in the current study. Previous studies showed that the gene expression and maturation or secretion of Cystatin C is controlled by transcriptional regulation and post-translational regulation [66]. Since the dimer formation of Cystatin C, which is an active form, could be prevented by inhibiting mitochondrial activity such as mitochondrial ROS (mtROS) generation [67], NOX4-dependent mtROS production might be a molecular mechanism of Cystatin C production in the hippocampus. In further study, we need to further investigate the interaction between NOX4 and the regulation of Cystatin C production in the hippocampus under hippocampal neuronal cells.

In our study, our results demonstrate that NOX4 plays an important role in the neuronal protection via Cystatin C-dependent hippocampal neurogenesis under chronic HFD. Additionally, NOX4 might be an important gene for the maintenance of hippocampal function during human metabolic diseases. Furthermore, our results suggest that the mutation of NOX4 gene could cause the functional reduction of hippocampal memory and learning ability under human metabolic diseases.

## 5. Conclusions

Consequently, we found the role of NOX4 in the neuronal protection during chronic metabolic stress by HFD. Particularly, we showed that NOX4 contributes to hippocampal neurogenesis via Cystatin C production under HFD. Our results suggest that NOX4 could be a critical gene for protecting against neuronal disorder during chronic metabolic diseases including obesity and type 2 diabetes.

## Figures and Tables

**Figure 1 genes-11-00567-f001:**
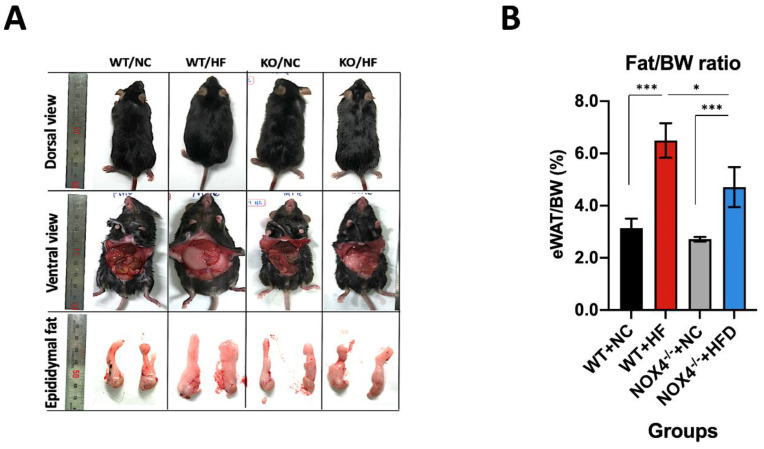
NADPH oxidase 4 (NOX4) deficiency decreases fat accumulation in epididymal white adipose tissue (eWAT) by high fat diet (HFD). (**A**) Representative images for dorsal view, ventral view, and epididymal fat from *Nox4*^−/−^ and wild-type (WT) mice with HFD or normal chow (NC) diet (*n* = 4 / group). Scale bars were indicated in each image. WT/NC; WT mice with normal chow (NC) diet, WT/HF; WT mice with high fat (HF) diet, KO/NC; *Nox4*^−/−^ mice with normal chow (NC) diet, KO/HF; *Nox4*^−/−^ mice with high fat (HF) diet. (**B**) Quantification of the ratio between epididymal white adipose tissue (eWAT) / body weight (BW) from *Nox4*^−/−^ and WT mice with HFD or NC diet (*n* = 4 / group). Data are mean ± standard deviation (SD). ***, *p* < 0.0005, *, *p* < 0.05, by Student’s two-tailed *t*-test.

**Figure 2 genes-11-00567-f002:**
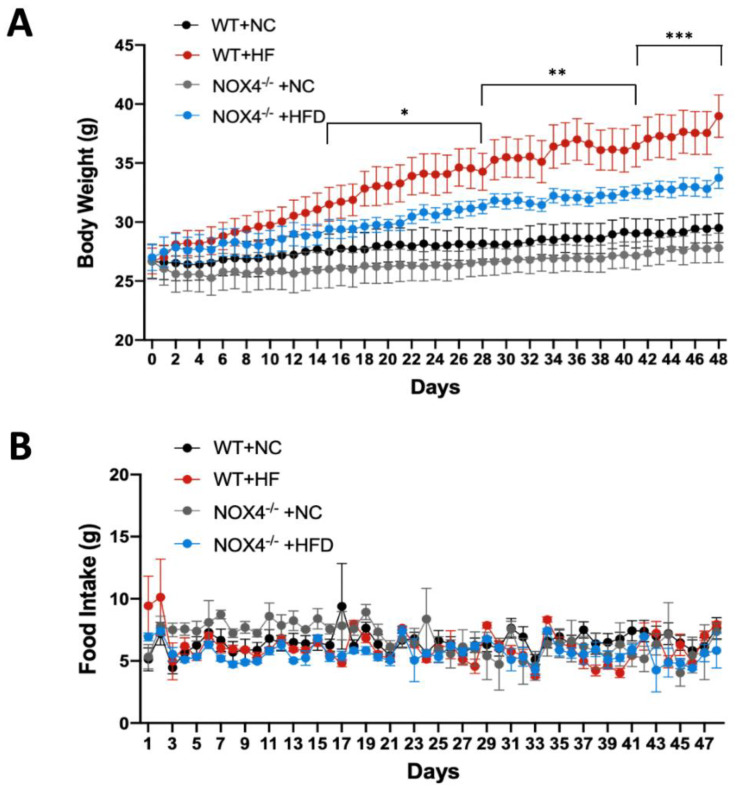
NOX4 deficiency contributes to the alteration of systemic metabolism under HFD. (**A**) Measurement of body weight from *Nox4*^−/−^ and wild-type (WT) mice with HFD or NC diet (*n* = 4 / group). Body weight was measured at same time in every day for seven weeks. (**B**) Measurement of food intake from *Nox4*^−/−^ and WT mice with HFD or NC diet (*n* = 4 / group). Food intake was measured at the same time every day for seven weeks. Data are mean ± standard deviation (SD). ***, *p* < 0.001, **, *p* < 0.005, *, *p* < 0.05 by Student’s two-tailed *t*-test.

**Figure 3 genes-11-00567-f003:**
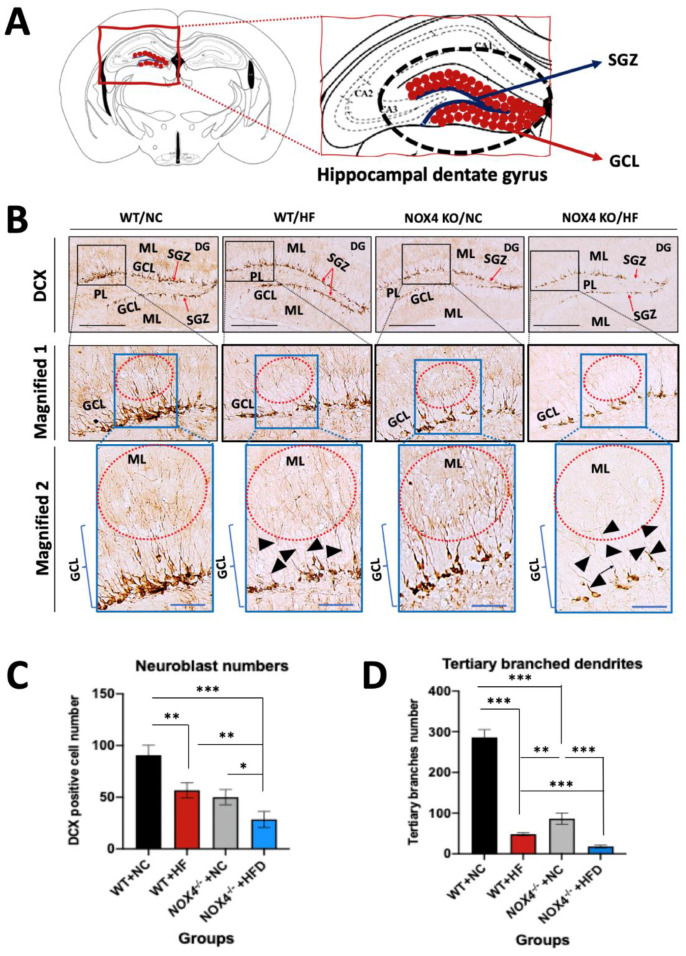
NOX4 deficiency decreases hippocampal neurogenesis by inhibiting neuronal maturation under HFD. (**A**) A diagram for the location of the doublecortin (DCX)-positive cells in the SGZ of hippocampal DG, which is an area between −1.82 and −2.46 mm posterior to the Bregma at the SGZ. (**B**) Representative immunohistochemistry image of DCX-positive cells by DCX staining in SGZ of hippocampal DG (red arrows) of brain tissues from *Nox4*^−/−^ and WT mice with HFD or NC diet. Positive area and cells are indicated by a red circle. In the low magnification image (top, DCX), scale bars are 200 μm. DCX-positive cells in SGZ were marked on a ‘black rectangle’ and drawings that magnify those cells in the SGZ can be observed in the ‘Magnified 1’. High magnification of the ‘blue rectangle’ of the ‘Magnified 1’ is shown in the ‘Magnified 2’. In the high magnification image (bottom, Magnified 2) scale, bars are 50 μm. Results are representative of each group (*n* = 4 / group). (**C**) Quantification of the numbers of DCX-positive cells (neuroblasts) in SGZ of hippocampal DG (red arrows) of brain tissues from *Nox4*^−/−^ and WT mice with HFD or NC. (**D**) Quantification of the numbers of tertiary branched dendrites in SGZ of hippocampal DG (red arrows) of brain tissues from *Nox4*^−/−^ and WT mice with HFD or NC. Data are mean ± standard deviation (SD). ***, *p* < 0.001, **, *p* < 0.005, *, *p* < 0.05 by Student’s two-tailed *t*-test. SGZ; adjacent sub-granular zone, ML; outer and middle molecular layers of the fascia dentata, GCL; granular cell layer, DG; dentate gyrus, PL; polymorphic layer, CA1; cornu ammonis area 1, CA2; cornu ammonis area 2, CA3; cornu ammonis area 3.

**Figure 4 genes-11-00567-f004:**
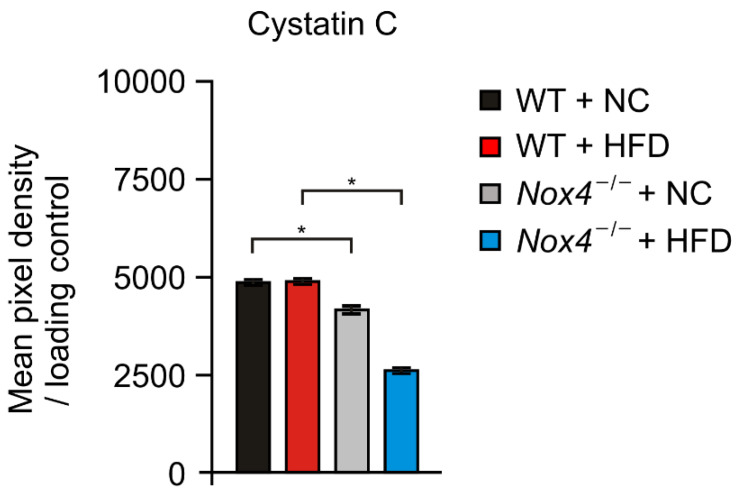
NOX4 deficiency suppresses the production of Cystatin C in the hippocampus during chronic HFD. Quantification of Cystatin C (left) secretion in the hippocampus from *Nox4*^−/−^ and WT mice with HFD or NC diet. Data are mean ± standard deviation (SD). *, *p* < 0.05 by Student’s two-tailed *t*-test.

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
