# Peer review of "NOX4 Deficiency Exacerbates the Impairment of Cystatin C-Dependent Hippocampal Neurogenesis by a Chronic High Fat Diet"

_genes, 2020, doi:10.3390/genes11050567_

Round 1

Reviewer 1 Report

This study investigates the effects of NOX4 deficiency on hippocampus neurogenesis and cystatine C levels in mice with normal and high fat diets. The neurogenesis decreased significantly in NOX4 knockout mice: not only the number of neuroblasts but also the dendritic development are inhibited. In the same time, the cystatine C level in the hippocampus decreased significantly. The NOX4 deficiency alone could cause the above alterations - the high fat diet exacerbated these alterations significantly. There are some flaws in the presented text and the experiments. INTRODUCTION: it is not clear which are the key factors in the regulation of adult neurogenesis in the hippocampus. The authors write about "metabolic" role of the hippocampus (lines 50-59) but the issues of the diabetes and obesity are not clear: how do these diseases affect hippocampus neurogenesis or the cognitive functions? There is no mention about the cellular localization of NOX4 in the brain, although there are literature data about it (Casas et al, 2017). METHODS: the NOX4 KO mice have to be precisely described. How is this NOX4 KO condition validated? Analysis of fat volume: obesity is clearly proved (Fig. 2). We do not understand "epididymal fat". Epididymis is an organ related to the testis. We think that the authors think "epidermal" fat. But we think that this dissected adipose tissue is not a precise measure of obesity because the method of the dissection is not described. Cystatine C and cytokine analysis: we think that cystatine C analysis is important, but the description of the method is lacking. The authors must describe the method of cystatine C determination (the supplementary material contains a picture, but there is no information about the assay). RESULTS: figures are appropriate. As to Fig. 4, we think that only cystatine C results are needed, because the others are not discussed and not relevant. We think that Fig. 5 is not necessary, because it is not giving us new information compared to the text (the same information is described in the text). DISCUSSION: not too relevant, because it is not presenting data about the relation of NOX4 and cystatine C. There is no effort to discuss the molecular mechanisms of NOX4 in relation of neurogenesis, no effort to discuss the mechanisms of cystatine C role in the neurogenesis. The relevant literature sources describe the importance of cystatine C in neuroprotection /Zou et al, 2017, Nature) and the effect of NOX4 deletion (Ma et al, 2018). The authors did not present ideas about the connection between NOX4 deletion and cystatine C decrease or down regulation. Conclusions: missing. List of abbreviations: missing. We think that obesity, NOX4 deficiency and cystatine C down regulation are three different phenomena. Reading this article we did not get new information about these phenomena, neither about the possible connections between them.

Author Response

Response to Genes Reviewer 1 Comments

This study investigates the effects of NOX4 deficiency on hippocampus neurogenesis and cystatine C levels in mice with normal and high fat diets. The neurogenesis decreased significantly in NOX4 knockout mice: not only the number of neuroblasts but also the dendritic development are inhibited. In the same time, the cystatine C level in the hippocampus decreased significantly. The NOX4 deficiency alone could cause the above alterations - the high fat diet exacerbated these alterations significantly. There are some flaws in the presented text and the experiments.

Reviewer’s Comment 1

INTRODUCTION: it is not clear which are the key factors in the regulation of adult neurogenesis in the hippocampus. The authors write about "metabolic" role of the hippocampus (lines 50-59) but the issues of the diabetes and obesity are not clear: how do these diseases affect hippocampus neurogenesis or the cognitive functions?

Response 1

As reviewer’s comment, we provided additional information and references for the relationship between the diabetes or obesity and hippocampus neurogenesis or the cognitive functions in Introduction section.

The following text has been added to Page 2. Line 56:

Page 2. Line 56 “Hippocampal neurogenesis is suppressed by the increase of oxidative stress and the decrease of brain-derived neurotrophic factor (BDNF) which is involved in the enhanced hippocampal neurogenesis under high fat diet (HFD) related to obesity and Type 2 diabetes [13-15]. Obesity is linked to in the development of mild cognitive impairment and late-life dementia or Alzheimer's disease [16-18]. Also, diabetes impairs hippocampus-dependent memory, perforant path synaptic plasticity and adult neurogenesis through the reduction of hippocampal function by glucocorticoid-mediated effects on new and mature neurons [19].”

Reviewer’s Comment 2

There is no mention about the cellular localization of NOX4 in the brain, although there are literature data about it (Casas et al, 2017).

Response 2

As reviewer’s comment, we provided the information of the cellular localization of NOX4 in the brain using previous paper by Casas et al, 2017 in Introduction section.

The following text has been added to Page 2. Line 75:

Page 2. Line 75 “In the brain, NOX4 is predominantly localized in endothelial cells and neurons in mouse and human [31,32].”

Reviewer’s Comment 3

METHODS: the NOX4 KO mice have to be precisely described. How is this NOX4 KO condition validated?

Response 3

As reviewer’s comment, we provided the information of NOX4 knockout mice in Materials and Methods section.

The following text has been added to Page 3. Line 99:

Page 3. Line 99 “The Nox4−/− mice were from the Jackson laboratory (Jackson laboratory, CT, USA). Nox4−/− mice were previously described [45]. The Nox4−/− mice were genotyped using standard PCR using DNA from tail.”

Reviewer’s Comment 4

Analysis of fat volume: obesity is clearly proved (Fig. 2). We do not understand "epididymal fat". Epididymis is an organ related to the testis. We think that the authors think "epidermal" fat. But we think that this dissected adipose tissue is not a precise measure of obesity because the method of the dissection is not described.

Response 4

As reviewer’s comment, we provided the more detail description for the utilization and analysis of epididymal white adipose tissues (eWAT) in Materials and Methods section.

The following text has been added to Page 3. Line 116:

Page 3. Line 116 “Mice (4 mice per group) were performed perfusion in each group. Epididymal white adipose tissues (eWAT) were used for the analysis of changes of fat mass by wight gain and cellularity measurement as described in the previous study [46,47]. Epididymal fat pads were carefully removed without damaging the testicular blood supply as previously described [48,49]. eWAT were completely separated from the epididymal fat pads and were calculated in each mouse before sacrifice.”

Reviewer’s Comment 5

Cystatine C and cytokine analysis: we think that cystatine C analysis is important, but the description of the method is lacking. The authors must describe the method of cystatine C determination (the supplementary material contains a picture, but there is no information about the assay).

Response 5

As reviewer’s comment, we provided the more detail description for the analysis of Cystatine C in Materials and Methods section.

The following text has been added to Page 4. Line 153:

Page 4. Line 153 “The protein lysates from hippocampus were incubated with 4 nitrocellulose membranes containing 111 different capture antibodies printed in duplicate for 16 h at 4 °C. Nitrocellulose membranes were incubated with detection antibody diluted in assay buffer for 2 h at room temperature and then incubated with the streptavidin-horseradish peroxidase (HRP) in assay buffer for 0.5 h at room temperature. The immunoreactive spots on nitrocellulose membranes were detected by the chemi reagent mix and then exposed to X-ray film. Multiple exposure times were used. The pixel densities from positive signals on developed X-ray film were collected and analyzed using a transmission mode scanner and image analysis software (HLImage++ Version 25.0.0r, https://www.wvision.com/QuickSpots.html (Western Vision Software, USA). The pixel densities were quantified and determined the relative change in analyte levels. The values of all analytes are shown in Supplemental table 1.”

Reviewer’s Comment 6

RESULTS: figures are appropriate. As to Fig. 4, we think that only cystatine C results are needed, because the others are not discussed and not relevant.

Response 6

As reviewer’s comment, we provided the only cystatine C results in Fig. 4 in Results section. We removed the others in Fig. 4. We moved the levels of others to Supplemental figure 2 and described in figure legend.

The following text has been added to Page 8. Line 261:

Page 8. Line 261 “On the other hand, FGF-1, a growth factor and signaling protein encoded by the FGF1 gene, and IL-28B, a cytokine encoded by the interferon lamda 3 (IFNL3) gene, levels in hippocampus were changed in Nox4-/- and WT with HFD or NC, whereas these changes were not co-related with hippocampal neurogenesis in Nox4-/- and WT mice (Figure S2 and Table S1).”

Reviewer’s Comment 7

We think that Fig. 5 is not necessary, because it is not giving us new information compared to the text (the same information is described in the text).

Response 7

As reviewer’s comment, we removed Fig. 5 in Results section.

Reviewer’s Comment 8

DISCUSSION: not too relevant, because it is not presenting data about the relation of NOX4 and cystatine C. There is no effort to discuss the molecular mechanisms of NOX4 in relation of neurogenesis, no effort to discuss the mechanisms of cystatine C role in the neurogenesis. The relevant literature sources describe the importance of cystatine C in neuroprotection /Zou et al, 2017, Nature) and the effect of NOX4 deletion (Ma et al, 2018). The authors did not present ideas about the connection between NOX4 deletion and cystatine C decrease or down regulation.

Response 8

As reviewer’s comment, we tried to discuss the molecular mechanisms between NOX4 and Cystatin C in the regulation of hippocampal neurogenesis in Discussion section. We provided the more detail description for the molecular mechanisms between NOX4 and Cystatin C.

The following text has been added to Page 9. Line 290 and Line 308:

Page 9. Line 290 “Our results show that the number of neuroblast (DCX)-positive cells in Nox4-/- mice under HFD as well as dendrites stretching out from those neuroblasts was significantly reduced compared to WT mice under HFD as well as Nox4-/- and WT mice under NC. Our findings suggest that NOX4 deficiency exacerbates the impairment of hippocampal neurogenesis under HFD. Our results are thought to be consistent with previous reports that a reduction in subventricular zone (SVZ) of lateral ventricular neurogenesis in NOX4 deficient condition [55,56]. Similarly, previous study showed that NOX4 deficiency contributes to the reduction of bone formation by osteogenic impairment [57]. Our findings suggest that NOX4 deficiency exacerbates the impairment of hippocampal neurogenesis via metabolic stress by HFD.”

Page 9. Line 308 “Cystatin C plays a neuroprotective role in neurodegenerative diseases including Parkin’s diseases (PD) Alzheimer’s diseases (AD), amyotrophic lateral sclerosis (ALS) and subarachnoid hemorrhage (SAH) [58-60]. In neurogenesis, the role of Cystatin C is identified in hippocampus [61,62]. The Cystatin C levels are co-related with the prominent neurogenesis [63,64]. Also, the basal level of neurogenesis in the subgranular layer of dentate gyrus are reduced in Cystatin C knockout mice, supporting a role for Cystatin C in neurogenesis [63,64]. During acute hippocampal injury or status epilepticus induced epileptogenesis, the Cystatin C gene and protein expression levels are increased in the hippocampus and in the dentate gyrus [65]. Also, previous study reported that Cystatin C has an important role in the generation of neuronal stem cells from embryonic stem (ES) cells [66]. Consistent with our results, there is an interaction between the levels of hippocampal neuronal differentiation and Cystatin C levels in the hippocampus at the NOX4 deficient mice. Our results showed that NOX4 deficiency exacerbates the reduction of Cystatin C production in hippocampus under chronic HFD compared to WT. These results suggest that the reduction of Cystatin C could be a critical molecular mechanism in the impairment of hippocampal neurogenesis by NOX4 deficiency under chronic HFD. Also, our results suggest that NOX4 could have an important role in the neuronal protection through hippocampal neurogenesis under human metabolic diseases such as obesity and type 2 diabetes.

Although our results showed the role of NOX4 in hippocampal neurogenesis and Cystatin C production under chronic HFD, there is limitation to determine the direct or indirect interaction between NOX4 and the regulation of Cystatin C production in hippocampal neurogenesis in current our study. Previous studies showed that the gene expression and maturation or secretion of Cystatin C is controlled by transcriptional regulation and post-translational regulation [67]. Since the dimer formation of Cystatin C, which is an active form, could be prevented by inhibiting mitochondrial activity such as mitochondrial ROS (mtROS) generation [67], NOX4-dependent mtROS production might be a molecular mechanism of Cystatin C production in hippocampus. In further study, we need the further investigation the interaction between NOX4 and the regulation of Cystatin C production in hippocampus under hippocampal neuronal cells.

In our study, our results demonstrate that NOX4 plays an important role in the neuronal protection via Cystatin C-dependent hippocampal neurogenesis under chronic HFD. Also, NOX4 might be an important gene for the maintenance of hippocampal function during human metabolic diseases. Furthermore, our results suggest that the mutation of NOX4 gene could cause the functional reduction of hippocampal memory and learning ability under human metabolic diseases.

Reviewer’s Comment 9

Conclusions: missing.

Response 9

As reviewer’s comment, we provided our conclusion in Conclusions Section.

The following text has been added to Page 10. Line 339:

Page 10. Line 339 “Consequently, we found that the role of NOX4 in the neuronal protection during chronic metabolic stress by HFD. Particularly, we showed that NOX4 contributes to hippocampal neurogenesis via Cystatin C production under HFD. Our results suggest that NOX4 could be a critical gene for the protection against neuronal disorder during chronic metabolic diseases including obesity and type 2 diabetes.”

Reviewer’s Comment 10

List of abbreviations: missing.

Response 10

As reviewer’s comment, we provided list of abbreviations.

The following text has been added to Page 10. Line 356.

Page 10. Line 356

“Abbreviations

NOX4 NADPH oxidase 4

NC Normal chow

HFD high fat diet

DCX doublecortin

GCL granular cell layer

DG dentate gyrus

SGZ subgranular zone

BDNF brain-derived neurotrophic factor

ROS reactive oxygen species

eWAT epidydimal white adipose tissue

mtROS mitochondrial ROS

ES embryonic stem”

Reviewer’s Comment 11

We think that obesity, NOX4 deficiency and cystatine C down regulation are three different phenomena. Reading this article we did not get new information about these phenomena, neither about the possible connections between them.

Response 11

As reviewer’s comment, we tried to demonstrate the molecular interaction among NOX4, Cystatin C and chronic metabolic stress by HFD such as obesity in our study. We believe that NOX4-dependent Cystatin C regulation in hippocampus could be a critical molecular event for the neuronal protection under human metabolic diseases including obesity and type 2 diabetes.

Reviewer 2 Report

The research article entitled “NOX4 deficiency exacerbates the impairment of Cystatin c-dependent hippocampal neurogenesis by chronic high fat diet” reports how NOX4 deficiency exacerbates the impairment of hippocampal neurogenesis by chronic high fat diet (HFD). Authors suggest that a corelated change in Cystatin c level could mediate the NOX4 deficiency. There are some interesting observations in this study specially related to high fat diet. High fat diet in wild type animals decreased new-born neuron count as well as their maturation. However, high fat diet in NOX4 knock-out showed decrease of new-born neuron count, similar to wild type, but accompanied with a severe deficiency in dendritic maturation. Overall, this is a well written and scientifically conducted study that merits publication in the journal. I would suggest following dimensions to authors that will help improvise the article:

  1. Nox4 is reported to alter proliferation, differentiation and maturation of new-born neurons. Authors claim “Here, we show that NOX4 deficiency exacerbates the impairment of hippocampal neurogenesis via inhibition of neuronal differentiation by chronic high fat diet (HFD).” This needs clarification because I believe that supportive evidence is lacking. BRDU pulse-chase experiments can answer this.

  1. Page 8, line 234: “We analyzed the levels of 111 soluble proteins including cytokines, chemokine and growth factors in hippocampus from Nox4-/- and WT mice with NC or HFD”. It will be extremely useful to have a table with all the soluble protein evaluated, may be as a heatmap (if not, then as a table in supplementary). This will give a better understanding to the reader about different aspects of the signaling mechanism underlying the changes reported by authors.

  1. Figure 5: The diagram shows that Cystatin c mediates the NOX4 effects. However, the evidence provided is just correlated decrease in the levels of Cystatin c. How did the correlated change can be presumed to be causal? Throughout the article including abstract presents the same without much experimental evidence.

  1. Abstract: “While the role of NADPH oxidase 4(NOX4) plays a role in the brain,”. The statement sounds absurd. Please reword it.

  1. There are several typos. Please edit and revise the article. For example, in abstract: “the mechanism by which NOX4 regulates hippocampal neurogenesis under metabolic stress unclear”. The line should read like “the mechanism by which NOX4 regulates hippocampal neurogenesis under metabolic stress is unclear”.

  1. Line no. 213: “NOX4 deficient dendrites have disconnected and shorten branches in the middle region compared to WT dendrites”. Please reword this sentence.

  1. Line no. 263: “changes in brain function depending on the type of food with the westernization of diet worldwide”. The choice of word “westernization of diet” does not seem scientific. Please be specific.

Author Response

Response to Genes Reviewer 2 Comments

The research article entitled “NOX4 deficiency exacerbates the impairment of Cystatin c-dependent hippocampal neurogenesis by chronic high fat diet” reports how NOX4 deficiency exacerbates the impairment of hippocampal neurogenesis by chronic high fat diet (HFD). Authors suggest that a corelated change in Cystatin c level could mediate the NOX4 deficiency. There are some interesting observations in this study specially related to high fat diet. High fat diet in wild type animals decreased new-born neuron count as well as their maturation. However, high fat diet in NOX4 knock-out showed decrease of new-born neuron count, similar to wild type, but accompanied with a severe deficiency in dendritic maturation. Overall, this is a well written and scientifically conducted study that merits publication in the journal. I would suggest following dimensions to authors that will help improvise the article:

Reviewer’s Comment 1

Nox4 is reported to alter proliferation, differentiation and maturation of new-born neurons. Authors claim “Here, we show that NOX4 deficiency exacerbates the impairment of hippocampal neurogenesis via inhibition of neuronal differentiation by chronic high fat diet (HFD).” This needs clarification because I believe that supportive evidence is lacking. BRDU pulse-chase experiments can answer this.

Response 1

Thank you for your valuable comment. We agree with your opinion and the supportive evidence related to our findings in hippocampal neurogenesis. In our system, the administration of BrdU needs more several weeks to affect the hippocampal neuronal differentiation under chronic HFD. The administration of BrdU in the middle of HFD cannot maintain the healthy condition of mice. We could not use the administration of BrdU in our study. We believe that your valuable advice helps to improve our current paper. Since we have a limitation for supporting neuronal differentiation using the administration of BrdU in our study, we described that our findings showed that NOX4 deficiency exacerbates the impairment of hippocampal neurogenesis via inhibition of neuronal maturation by chronic high fat diet (HFD) in Abstract section following your comment. Also, we mentioned our limitation related to supporting neuronal differentiation in Discussion section.

The following text has been added to Page 1. Line 25 in Abstract section:

Page 1. Line 25 “Here, we show that NOX4 deficiency exacerbates the impairment of hippocampal neurogenesis via inhibition of neuronal maturation by chronic high fat diet (HFD).”

The following text has been added to Page 9. Line 297 in Discussion section:

Page 9. Line 297 “Our findings suggest that NOX4 deficiency exacerbates the impairment of hippocampal neurogenesis via metabolic stress by HFD. Although we showed that the role of NOX4 in hippocampal neurogenesis during HFD, there is a limitation to support whether NOX4 can directly affect neuronal differentiation in hippocampal neurogenesis in our current study. Further study for the role of NOX4 on hippocampal neuronal differentiation would need to be studied.”

Reviewer’s Comment 2

Page 8, line 234: “We analyzed the levels of 111 soluble proteins including cytokines, chemokine and growth factors in hippocampus from Nox4-/- and WT mice with NC or HFD”. It will be extremely useful to have a table with all the soluble protein evaluated, may be as a heatmap (if not, then as a table in supplementary). This will give a better understanding to the reader about different aspects of the signaling mechanism underlying the changes reported by authors.

Response 2

As reviewer’s comment, we provided the list and values all the soluble protein levels in Supplemental table 1 (Please see new Supplemental table 1) and more detail description for the analysis method of Cystatine C in Materials and Methods section.

The following text has been added to Page 4. Line 153:

Page 4. Line 153 “The protein lysates from hippocampus were incubated with 4 nitrocellulose membranes containing 111 different capture antibodies printed in duplicate for 16 h at 4 °C. Nitrocellulose membranes were incubated with detection antibody diluted in assay buffer for 2 h at room temperature and then incubated with the streptavidin-horseradish peroxidase (HRP) in assay buffer for 0.5 h at room temperature. The immunoreactive spots on nitrocellulose membranes were detected by the chemi reagent mix and then exposed to X-ray film. Multiple exposure times were used. The pixel densities from positive signals on developed X-ray film were collected and analyzed using a transmission mode scanner and image analysis software (HLImage++ Version 25.0.0r, https://www.wvision.com/QuickSpots.html (Western Vision Software, USA). The pixel densities were quantified and determined the relative change in analyte levels.”

The following text has been added to Page 8. Line 261:

Page 8. Line 261 “On the other hand, FGF-1, a growth factor and signaling protein encoded by the FGF1 gene, and IL-28B, a cytokine encoded by the interferon lamda 3 (IFNL3) gene, levels in hippocampus were changed in Nox4-/- and WT with HFD or NC, whereas these changes were not co-related with hippocampal neurogenesis in Nox4-/- and WT mice (Figure S2 and Table S1).”

Reviewer’s Comment 3

Figure 5: The diagram shows that Cystatin c mediates the NOX4 effects. However, the evidence provided is just correlated decrease in the levels of Cystatin c. How did the correlated change can be presumed to be causal? Throughout the article including abstract presents the same without much experimental evidence.

Response 3

As reviewer’s comment, we removed Fig. 5 in Results section.

Reviewer’s Comment 4

Abstract: “While the role of NADPH oxidase 4(NOX4) plays a role in the brain,”. The statement sounds absurd. Please reword it.

Response 4

As reviewer’s comment, we revised the current sentence in Abstract section.

The following text has been added to Page 1. Line 23:

Page 1. Line 23 “While the role of NADPH oxidase 4 (NOX4) plays in the brain, the mechanism by which NOX4 regulates hippocampal neurogenesis under metabolic stress is unclear.”

Reviewer’s Comment 5

There are several typos. Please edit and revise the article. For example, in abstract: “the mechanism by which NOX4 regulates hippocampal neurogenesis under metabolic stress unclear”. The line should read like “the mechanism by which NOX4 regulates hippocampal neurogenesis under metabolic stress is unclear”.

Response 5

As reviewer’s comment, we revised the current sentence in Abstract section.

The following text has been added to Page 1. Line 23:

Page 1. Line 23 “While the role of NADPH oxidase 4 (NOX4) plays in the brain, the mechanism by which NOX4 regulates hippocampal neurogenesis under metabolic stress is unclear.”

Reviewer’s Comment 6

Line no. 213: “NOX4 deficient dendrites have disconnected and shorten branches in the middle region compared to WT dendrites”. Please reword this sentence.

Response 6

As reviewer’s comment, we revised the current sentence in Results section at Line 232.

The following text has been added to Page 7. Line 232:

Page 7. Line 232 “Nox4-/- mice showed higher disconnected and short branched dendrites in the middle region of DG under HFD or NC than WT mice”

Reviewer’s Comment 7

Line no. 263: “changes in brain function depending on the type of food with the westernization of diet worldwide”. The choice of word “westernization of diet” does not seem scientific. Please be specific.

Response 7

As reviewer’s comment, we revised the current sentence in Discussion section at Line 278.

The following text has been added to Page 8. Line 278:

Page 8. Line 278 “Imbalanced foods contained with high fat or high calories has been implicated in the changes of brain function in human metabolic diseases such as obesity and type 2 diabetes [7-10].”

Round 2

Reviewer 1 Report

The critical objections of the Reviewer were all accepted and answered. This version of the manuscript is completely acceptable.

Author Response

Response to Genes Reviewer 1 Comments

Reviewer’s Comment 1

The critical objections of the Reviewer were all accepted and answered. This version of the manuscript is completely acceptable.

Response 1

Thank you so much for your valuable comments and time.

Reviewer 2 Report

Thank you for making the appropriate changes. The only suggestion that was not completely addressed is the following. Otherwise, the article should be acceptable in my view.

Page 7. Line 232 “Nox4-/- mice showed higher disconnected and short branched dendrites in the middle region of DG under HFD or NC than WT mice”

"Disconnected dendrites" is inappropriate. What do you mean by this ? Please remove this and related analysis unless there is a good explanation.

Author Response

Response to Genes Reviewer 2 Comments

Reviewer’s Comment 1

Thank you for making the appropriate changes. The only suggestion that was not completely addressed is the following. Otherwise, the article should be acceptable in my view.

Page 7. Line 232 “Nox4-/- mice showed higher disconnected and short branched dendrites in the middle region of DG under HFD or NC than WT mice”

"Disconnected dendrites" is inappropriate. What do you mean by this ? Please remove this and related analysis unless there is a good explanation.

Response 1

Thank you for your valuable comment. As reviewer’s comment, we understand your point. We revised the current sentence using ‘immature dendrites which have fewer and shorter branches’ in Results section at Line 232.

The following text has been added to Page 7. Line 232:

Page 7. Line 232 “Nox4-/- mice showed higher immature dendrites which have fewer and shorter branches in the middle region of DG under HFD or NC than WT mice”